# Comparison of the Blood–Brain Barrier Transport and Vulnerability to P-Glycoprotein-Mediated Drug–Drug Interaction of Domperidone versus Metoclopramide Assessed Using In Vitro Assay and PET Imaging

**DOI:** 10.3390/pharmaceutics14081658

**Published:** 2022-08-09

**Authors:** Louise Breuil, Sébastien Goutal, Solène Marie, Antonio Del Vecchio, Davide Audisio, Amélie Soyer, Maud Goislard, Wadad Saba, Nicolas Tournier, Fabien Caillé

**Affiliations:** 1Laboratoire d’Imagerie Biomédicale Multimodale (BIOMAPS), Université Paris-Saclay, CEA, CNRS, Inserm, Service Hospitalier Frédéric Joliot, 4 place du Général Leclerc, 91401 Orsay, France; 2Pharmacy Department, Robert-Debré Hospital, AP-HP, Université Paris Cité, 75019 Paris, France; 3Pharmacy Department, Bicêtre Hospital, AP-HP, Université Paris-Saclay, 94270 Le Kremlin-Bicêtre, France; 4CEA, Département Médicaments et Technologies pour la Santé, SCBM, Université Paris-Saclay, 91190 Gif-sur-Yvette, France

**Keywords:** ATP-binding cassette, drug–drug interaction, membrane transporter, neuropharmacology, pharmacokinetics, PET imaging, radiochemistry, P-glycoprotein, blood–brain barrier

## Abstract

Domperidone and metoclopramide are widely prescribed antiemetic drugs with distinct neurological side effects. The impact of P-glycoprotein (P-gp)-mediated efflux at the blood–brain barrier (BBB) on brain exposure and BBB permeation was compared in vitro and in vivo using positron emission tomography (PET) imaging in rats with the radiolabeled analogs [^11^C]domperidone and [^11^C]metoclopramide. In P-gp-overexpressing cells, the IC_50_ of tariquidar, a potent P-gp inhibitor, was drastically different using [^11^C]domperidone (221 nM [198–248 nM]) or [^11^C]metoclopramide (4 nM [2–8 nM]) as the substrate. Complete P-gp inhibition led to a 1.8-fold higher increase in the cellular uptake of [^11^C]domperidone compared with [^11^C]metoclopramide (*p* < 0.0001). Brain PET imaging revealed that the baseline brain exposure (AUC_brain_) of [^11^C]metoclopramide was 2.4-fold higher compared with [^11^C]domperidone (*p* < 0.001), consistent with a 1.8-fold higher BBB penetration (AUC_brain_/AUC_plasma_). The maximal increase in the brain exposure (2.9-fold, *p* < 0.0001) and BBB penetration (2.9-fold, *p* < 0.0001) of [^11^C]metoclopramide was achieved using 8 mg/kg of tariquidar. In comparison, neither 8 nor 15 mg/kg of tariquidar increased the brain exposure of [^11^C]domperidone (*p* > 0.05). Domperidone is an avid P-gp substrate that was in vitro compared with metoclopramide. Domperidone benefits from a lower brain exposure and a limited risk for P-gp-mediated drug–drug interaction involving P-gp inhibition at the BBB.

## 1. Introduction

Domperidone and metoclopramide are both dopamine D2 receptor antagonists prescribed as antiemetic drugs. Their pharmacological targets are the peripheral D2 receptors, especially at the area postrema level, a spinal structure responsible for the vomiting reflex. Metoclopramide is also a 5HT_3_ serotonin receptor antagonist responsible for its gastroprokinetic effect. Their action on the D2 receptors of the central nervous system (CNS) is related to adverse events. They are rarely observed with domperidone, which acts at the antehypophyse level, generating gynecomastia and galactorrhea [1]. The Food and Drug Administration (FDA) put a black-box warning on metoclopramide, thus restricting its clinical use [2]. Metoclopramide has been shown to cause neurological side effects such as peripheral extrapyramidal symptoms and tardive dyskinesia. The risks increase when metoclopramide is administered in high doses and during long-term treatment [3]. Table 1 shows the CNS relative frequencies of the side effects of domperidone and metoclopramide.

These differences may be explained by lower brain penetration of domperidone compared with metoclopramide [7]. It is, however, difficult to estimate brain penetration and exposure in vivo. There is a need for minimally invasive methods to explore and compare the brain penetration of such compounds with clinical perspectives.

Both domperidone and metoclopramide are substrates of the P-glycoprotein (P-gp, ABCB1), the main efflux transporter expressed at the blood–brain barrier (BBB). A large body of literature suggests that the P-gp function controls the brain exposure to many drugs [8]. The most widely accepted hypothesis for low brain penetration of domperidone is that domperidone is an avid substrate of the P-gp [9]. In contrast, metoclopramide is considered a comparatively “weak” P-gp substrate in vitro [10], implying that metoclopramide exerts CNS effects despite fully functional P-gp at the BBB.

Comparison of the brain kinetics of domperidone and metoclopramide can first be informed by available data in the literature. The BBB passage of domperidone and metoclopramide has been compared in an in vitro model of the BBB using bovine brain capillary endothelial cells grown on filter inserts in a co-culture system with glial cells [11]. Compared with the passage of solutes across the filter (no BBB), the authors reported a 76% passage for domperidone versus 100% for metoclopramide [11]. The transport across parallel artificial membrane permeability assays (PAMPA) was 76-fold lower for domperidone compared with metoclopramide, suggesting higher passive diffusion of metoclopramide across lipid membranes [12]. The efflux ratio of domperidone in cells overexpressing the human P-gp (MDCK-MDR1), using bidirectional transport assay, was 31.2 for domperidone. This transport was reduced to ~1 in the presence of P-gp inhibition, thus confirming that domperidone is an avid substrate of the P-gp [13]. In comparison, a 1.4 efflux ratio for metoclopramide was reported using the same MDCK-MDR1 cell line from the same origin, suggesting that metoclopramide is a much “weaker” substrate of the P-gp [10]. This was also observed in cells transfected with the rodent P-gp gene (LLC-PK1-Mdr1a), in which the efflux ratio of domperidone was 87.5 versus 1.6 for metoclopramide [7]. Ex vivo studies in rats reported a tissue/plasma ratio (K_p,brain_) of 0.17 for domperidone [14] versus 1.72 for metoclopramide [15]. The K_p,uu,brain_, which takes the binding to plasma and brain proteins into account, was 0.022 for domperidone versus 2.4 for metoclopramide in rats [16]. Another rat study consistently reported a K_p,uu,brain_ of 0.044 for domperidone versus 0.235 for metoclopramide [7]. A 6.6-fold increase in the K_p,brain_ of metoclopramide was observed in P-gp deficient mice compared with wild-type mice [17]. Schinkel et al. reported that the K_p,brain_ of radiolabeled domperidone (^3^H-domperidone) was not increased in P-gp deficient mice, which was attributed to the predominant proportion of circulating radiometabolites, which rapidly surpassed the radioactive signal of parent unmetabolized ^3^H-domperidone in plasma [9]. However, P-gp deficiency was associated with increased CNS effects and catalepsy compared with wild-type mice when an oral pharmacological dose of domperidone was administered. This suggests a higher level of domperidone in the CNS of P-gp-deficient mice. However, as P-gp is abundant in the intestinal epithelium, increased oral bioavailability of domperidone in P-gp-deficient mice may also have contributed to the observed effect [9,18]. By using in situ brain perfusion, which bypasses the effect of plasma protein binding and peripheral metabolism, a significant increase of the K_p,brain_ of domperidone was observed in the presence of P-gp inhibition achieved using high concentration of the P-gp inhibitors cyclosporin A (2.5 µM) [19] or verapamil (500 µM) in the perfusate, which could hardly be achieved in vivo.

It is broadly assumed that avid substrates expose to a higher risk of drug–drug interactions (DDIs) caused by P-gp inhibitors. In comparison, weak substrates have less impact on brain kinetics, assuming a limited risk for clinically relevant DDI [20]. It may therefore be hypothesized that the consequences of DDI precipitated by P-gp inhibitors may lead to unintended brain exposure and CNS effects, which may balance the CNS safety of domperidone in this particular situation.

Positron emission tomography (PET) imaging provides an advanced technology to study in vivo the crossing of biological barriers, as it allows for the study of drug distribution and pharmacokinetics in a non-invasive way. Both metoclopramide and domperidone can be isotopically radiolabeled with carbon-11, i.e., without modification of the chemical structure and, by extension, the biological properties of the drug. [^11^C]metoclopramide has been validated as a PET probe for detecting the regulation of P-gp function at the BBB in animals and humans, including the impact of inhibition and induction of the transporter [21,22,23]. Accumulated in vitro and in vivo data suggest the high vulnerability of [^11^C]metoclopramide to P-gp inhibition. Availability of [^11^C]domperidone for PET imaging offers the unique opportunity to compare the brain kinetics of domperidone with previously published data of metoclopramide [21] obtained using the same approach [24]. This study assesses the P-gp substrate properties of domperidone in vitro in transfected cells expressing the human P-gp. Baseline brain exposure and consequences of DDI precipitated by tariquidar (TQD), a potent inhibitor of P-gp, are then assessed in vivo using PET imaging in rats.

## 2. Materials and Methods

### 2.1. [^11^C]Metoclopramide PET Data

All in vitro and in vivo experiments performed using [^11^C]metoclopramide presented in this article have already been published [21,24]. However, some newly estimated pharmacokinetic parameters and previously published data relative to [^11^C]metoclopramide brain PET imaging are displayed to allow for direct comparison with data obtained using [^11^C]domperidone.

### 2.2. Chemistry

Tariquidar (TQD) used for P-gp inhibition was purchased from Eras Labo (Saint-Nazaire-les-Eymes, France). TQD solutions for intravenous injections were freshly prepared at the selected concentrations on the day of the experiment by dissolving TQD dimesylate 2.35 H_2_O in dextrose solution (5%, *w*/*v*), followed by dilution with sterile water. *O*-desmethyl metoclopramide was purchased from LGC standards (France). The precursor for [^11^C]domperidone radiosynthesis was synthesized in house [25].

### 2.3. Radiochemistry

[^11^C]metoclopramide was synthesized by automated radiomethylation using a TRACERlab^®^ FX C Pro module (GE Healthcare, Buc, France) and cyclotron-produced [^11^C]CO_2_ according to the method described in the literature [26]. No carrier added [^11^C]CO_2_ (50–70 GBq) was produced via the ^14^N(p, α)^11^C nuclear reaction by irradiation of an [^14^N]N_2_ target containing 0.15–0.5% of O_2_ on a cyclone 18/9 cyclotron (18 MeV, IBA, Ottignies-Louvain-la-Neuve, Belgium). [^11^C]CO_2_ was subsequently reduced to [^11^C]CH_4_ and iodinated to [^11^C]CH_3_I following the process described by Larsen et al. [27] and finally converted to [^11^C]CH_3_OTf according to the method of Jewett [28]. [^11^C]CH_3_OTf was bubbled into a solution of *O*-desmethyl-metoclopramide (1 mg) and aqueous sodium hydroxide (3 M, 7 μL) in acetone (400 μL) at −20 °C for 3 min. The mixture was heated at 110 °C for 2 min, then evaporating the residual solvent to dryness at 110 °C under a vacuum for 30 s. Upon cooling to 60 °C, a mixture of aqueous NaH_2_PO_4_ (20 mM)/CH_3_CN/H_3_PO_4_ (85/15/0.2 *v*/*v*/*v*) was added. Purification was realized by reverse phase HPLC (Waters Symmetry^®^ C18 7.8 × 300 mm, 7 μm) with a 501 HPLC Pump (Waters, Guyancourt, France) using aqueous NaH_2_PO_4_ (20 mM)/CH_3_CN/H_3_PO_4_ (85/15/0.2 *v*/*v*/*v*, 5 mL/min) as eluent. UV detection (K2501, Knauer, Germany) was performed at 220 nm. The purified compound was diluted with water (20 mL) and passed through a Sep-Pak^®^ C18 cartridge (Waters, Milford, CT, USA). The cartridge was rinsed with water (10 mL) and eluted with ethanol (2 mL). The final compound was diluted with saline (0.9% *w*/*v*, 8 mL) to afford ready-to-inject [^11^C]metoclopramide (1.9 ± 0.2 GBq) in 12% ± 3% radiochemical yield (RCY) within 40 min and with a molar activity (MA) of 88 ± 13 GBq/μmol at the end of the beam (EOB) (*n* = 35) (Figure 1A).

[^11^C]Domperidone was synthesized by radiocarbonylation following the original Staudiger Aza-Wittig (SAW) method described in the literature [25]. Automated radiolabelling was performed using a MeIplus research module (Synthra GmbH, Hamburg, Germany) with modifications to undergo direct bubbling of the [^11^C]CO_2_. No carrier-added [^11^C]CO_2_ (30–40 GBq) was produced via the ^14^N(p, α)^11^C nuclear reaction by irradiation of an [^14^N]N_2_ target containing 0.15–0.5% of O_2_ on a cyclone 18/9 cyclotron (18 MeV, IBA, Ottignies-Louvain-la-Neuve, Belgium) and trapped at −180 °C. [^11^C]CO_2_ was then released at 0 °C under a stream of helium (8 mL/min) to bubble for 10 s into the reaction vessel containing a solution of the precursor (1 mg) and dimethylphenyl phosphine (15 µL) in DMF (300 µL) at −50 °C. The mixture was heated at 20 °C for 5 min and hydrolyzed with glacial acetic acid (100 µL), followed by a mixture of H_2_O/CH_3_CN/TFA (65/35/0.1 *v*/*v*/*v*, 1 mL). The crude product was purified by semi-preparative HPLC on a reverse-phase Symmetry C18 column (250 × 4.6 mm, 5 μm, Waters, Milford, CT, USA) using a mixture of H_2_O/CH_3_CN/TFA (65/35/0.1 *v*/*v*/*v*, 5 mL/min) as eluent with gamma and UV (λ = 280 nm) detection. The collected peak of [^11^C]domperidone was diluted with water (20 mL) and loaded on a C18 cartridge (Sep-Pak C18, Waters, Milford, CT, USA). The cartridge was rinsed with water (10 mL), and the product was eluted with ethanol (2 mL) and further diluted with aq. 0.9% NaCl (8 mL). Ready-to-inject [^11^C]domperidone (5.9 ± 0.3 GBq) was obtained within 30 min from the end of the beam in 47 ± 4% RCY and 85 ± 15 GBq/µmol MA at the EOB (*n* = 12) (Figure 1B).

For both radiotracers, quality control was performed by HPLC using a 717 plus Autosampler system equipped with a 1525 binary pump and a 2996 photodiode array detector (Waters, Milford, CT, USA) and a Flowstar LB 513 (Berthold, Thoiry, France) gamma detector. The system was monitored with the Empower 3 (Waters, Milford, CT, USA) software. HPLC were realized on a reverse-phase analytical Symmetry C18 (50 × 3.9 mm, 5 μm, Waters, Milford, CT, USA) column using either a mixture of aqueous NaH_2_PO_4_ (4 mM)/acetonitrile/H_3_PO_4_ (90/10/0.2 *v*/*v*/*v*, 2 mL/min) or a mixture of H_2_O/CH_3_CN/PicB7^®^ (70/30/0.2 *v*/*v*/*v*, 2 mL/min) as eluent for [^11^C]metoclopramide or [^11^C]domperidone, respectively. UV detection was performed at 274 nm or 285 nm for [^11^C]metoclopramide or [^11^C]domperidone, respectively. Identification of the peak was assessed by comparing the retention time of the carbon-11-labeled compound with the retention time of the non-radioactive reference (t_Rref_). For acceptance, the retention time must be within the t_Rref_ ±10% range. Radiochemical purity (RCP) was calculated as the ratio of the peak’s area under the curve (AUC) over the sum of the AUCs of all other peaks on gamma chromatograms. RCP is the mean value of three consecutive runs. The RCY of the labeling reaction was calculated as the ratio of the decay-corrected activity at the end of the synthesis (A_EOS_) CO_2_ measured in an ionization chamber (Capintec^®^, Berthold, Thoiry, France) over the starting activity of [^11^C]CO_2_ (A_CO2_) measured by the calibrated detector of the synthesizer. This ratio was corrected for the RCP following the equation: RCY = (A_EOS_/A_CO2_) × RCP. MA was calculated as the ratio of the activity of the collected peak of the radioactive product measured in an ionization chamber (Capintec^®^, Berthold, Thoiry, France) over the molar quantity of the compound determined using calibration curves. MA was calculated as the mean value of three consecutive runs.

Chemical characterization of [^11^C]metoclopramide and [^11^C]domperidone is reported in the Appendix A. The Appendix A shows HPLC and radioHPLC data for each radiotracer. 

### 2.4. Uptake Assay in P-gp Overexpressing Cells

Culture media and buffers were obtained from Fischer Scientific, France. Stably transfected MDCKII-MDR1 cells were obtained from Dr. Alfred Schinkel (National Cancer Institute, Amsterdam, The Netherlands) and were grown under a controlled atmosphere at 37 °C, 5% CO_2_. The culture medium was composed of DMEM Glutamax (Dulbecco’s Modified Eagle Medium, 4.5 g/Ldextrose, 1 mM pyruvate) supplemented with 10% fetal bovine serum and 1% antibiotics (penicillin and streptomycin 5000 U/mL).

P-gp-mediated transport of [^11^C]domperidone and [^11^C]metoclopramide was compared in cells expressing human P-gp. Cells were seeded in 24-well plates (30,000 cells per well) in a 500 μL culture medium. Cells were grown to confluence (~2 days). On the day of the experiment, the culture medium was removed and replaced by 200 μL of incubation buffer (10% HBSS (Hanks’ Balanced Salt Solution) + 1.26 mM CaCl_2_ + 0.49 mM MgCl_2_) containing 1 mM pyruvate and 10 mM HEPES (4-(2-hydroxyethyl)-1-piperazineethanesulfonic acid, 37 °C). The incubation buffer contained the tested radiolabeled P-gp substrate (~37 MBq/40 mL corresponding to <1 µg/40 mL of the unlabeled compound) and TQD at the selected concentration. TQD was dissolved in DMSO, and TQD concentrations ranged from 0 to 800 nM (1% *v*/*v* final DMSO concentration). After 30 min of incubation at 37 °C, the buffer was removed, and cell monolayers were rapidly washed with 300 μL of ice-cold Dulbecco’s phosphate buffer. Cells were then lysed with 500 μL of NaOH (10 mM, 10 min). Then, 400 μL of cell lysate was collected from each well (*n* = 4 wells per condition) and gamma counted using a Cobra Quantum (Perkin-Elmer, Villebon-sur-Yvette, France).

### 2.5. Animals

A total of 12 male Wistar rats (Janvier, Le Genest-Saint-Isle, France) (263 ± 34 g) were used to investigate the brain kinetics of [^11^C]domperidone. Animals were housed and acclimatized for at least 3 days before the experiments. Rats had free access to chow and water. All animal experiments were in accordance with the recommendations of the European Community (2010/63/UE) and the French National Committees (law 2013-^11^8) for the care and use of laboratory animals. The experimental protocol was approved by a local ethics committee for animal use (CETEA) and by the French Ministry of Agriculture (APAFIS#74662016110417049220 v2). The sample size for each group was based on previous studies [21,24].

### 2.6. PET Experiments

PET acquisitions were performed using an Inveon microPET scanner (Siemens, TN, USA). Anesthesia was induced and then maintained using 3.5% followed by 1.5–2.5% isoflurane in O_2_. Radiotracers were diluted in saline to obtain at least <10% ethanol in the ready-to-inject preparation. Thirty-minute dynamic scans were acquired, starting with an intravenous bolus injection of [^11^C]domperidone (32 ± 5 MBq, 0.80 ± 0.55 µg) via a catheter inserted in the caudal lateral vein. PET data of [^11^C]metoclopramide used for comparison with [^11^C]domperidone have already been reported. The injected dose of [^11^C]metoclopramide used for comparison was (35 ± 5 MBq, 3.4 ± 1.3 µg) [24].

### 2.7. P-gp Inhibition Protocol 

TQD was used to inhibit P-gp function in rats. TQD was injected in a volume of 200–300 µL into the caudal vein 15 min before radiotracer injection. PET acquisitions using [^11^C]domperidone were performed after vehicle injection, 8 mg/kg (*n* = 1) or 15 mg/kg (*n* = 4) of TQD.

### 2.8. Arterial Input Function 

Dedicated experiments were performed in additional rats to estimate the metabolism and arterial input function of [^11^C]domperidone in the absence and the presence of P-gp inhibition using TQD (*n* = 2 for each condition).

Blood samples (50 µL) were collected at selected times from the femoral artery to establish total radioactivity kinetics in arterial plasma. Plasma was separated from whole blood by centrifugation (5 min, 2054 g, 4 °C), and 20 µL of plasma and whole blood were counted using a PET cross-calibrated gamma well counter (WIZARD2, PerkinElmer, Villebon-sur-Yvette, France) to obtain the whole-blood and plasma activity curves. All data were corrected for radioactive decay from the injection time. For the larger blood samples, 80 µL plasma was deproteinized with acetonitrile. The supernatant was injected in high-performance liquid chromatography, equipped with an Atlantis^®^ T3 5 μm 10 × 250 mm column (Waters, Guyancourt, France) and an Atlantis^®^ T3 5 μm 19 × 10 mm pre-column (Waters) with an LB-514 radioactivity flow detector (Berthold, France, MX Z100 cell). The mobile phase was composed of water containing 0.1% (*v*/*v*) TFA (solvent A) and acetonitrile containing 0.1% (*v*/*v*) TFA (solvent B) delivered in a gradient elution mode at a flow rate of 5 mL min^−1^: solvent B increased linearly from 20 to 30% from 0 to 9 min. The [^11^C]domperidone parent fraction was calculated as a percentage of the total radioactivity (metabolites and parent).

For each animal, a 1-exponential decay function was fitted to [^11^C]domperidone parent fraction, which was time multiplied with the plasma activity curve to obtain the metabolite-corrected arterial plasma input function used for the kinetic modeling.

The method for [^11^C]metoclopramide metabolism has been previously reported [21].

### 2.9. Data Analysis and Statistics

#### 2.9.1. In Vitro Data

Counting values obtained in each well were normalized using Equation (1):(1)$$I\%=\frac{\left(R-mR_{zero}\right)}{mR_{max}}\times 100$$
where *I*% is the extent of inhibition (*I*%) between 0 and 100%, *R* is the radioactivity in the 4 tested wells, *mR_zero_* is the mean of the 4 wells without TQD, and *mR_max_* is the difference of means between the 4 wells containing the highest concentration of TQD (800 nM) and the 4 wells without TQD.

The in vitro half-maximum inhibitory concentration (IC_50_) was estimated by non-linear regression using the “One-site binding Hill equation” function in Graphpad Prism software (V8.0, San Diego, CA, USA) with maximal uptake constrained to 100%. IC_50_ values obtained for each radiotracer were considered different when their 95% confidence interval (CI_95%_) did not overlap. The uptake ratio (mean ± S.D) of each P-gp substrate was determined as the maximal intracellular radioactivity after complete inhibition of P-gp (800 nM TQD for [^11^C]-domperidone and 200 nM for [^11^C]metoclopramide) divided by the mean intracellular radioactivity without P-gp inhibition. The normality of the data was checked using the Kolmogorov–Smirnov test. Uptake ratios were then compared using the *t*-test. 

#### 2.9.2. PET Data Analysis

Images were reconstructed with the Fourier rebinning algorithm and the three-dimensional ordered-subset expectation-maximization algorithm, including normalization, attenuation, scatter, and random corrections. Image analysis and quantification of radioactivity uptake were performed using PMOD software (version 3.9; PMOD Technologies, Zurich, Switzerland). A region of interest was drawn over the whole brain to generate time-activity curves (TACs) with time frame durations of 0.25 min, 2 × 0.5 min, 0.75 min, 4 × 1 min, 1.5 min, 4 × 2 min, 3 × 2.5 min, 3 × 3 min, and 3.5 min. Brain radioactivity was corrected for ^11^C decay and expressed as the standardized uptake value (SUV) after correction by injected dose and animal weight.

Brain and plasma exposure of [^11^C]domperidone was estimated by the area under the TAC between 0 and 30 min and 20 and 30 min in the brain (AUC_brain 0–30_; AUC_brain 20–30_; *n* = 3–4 per condition) and between 20 and 30 min in the plasma (AUC_plasma 20–30_; *n* = 2 per condition). The mean AUC_plasma 20–30_ was obtained in each condition to estimate K_p,brain_ = AUC_brain 20–30_/AUC_plasma 20–30,_ considering the mean metabolite-corrected arterial input function. K_p,brain_ were compared with a two-way ANOVA and a Tukey’s post hoc test using Graphpad Prism software (V8.0, San Diego, CA, USA).

## 3. Results

### 3.1. Radiochemistry

Automated radiomethylation under standard conditions using the *O*-desmethyl precursor of metoclopramide afforded ready-to-inject [^11^C]metoclopramide (1.9 ± 0.2 GBq) in 12% ± 3% RCY within 40 min and with a MA of 88 ± 13 GBq/μmol (*n* = 35) (Figure 1A). Quality control revealed a chemical and RCP above 99%, making [^11^C]metoclopramide suitable for in vivo injection in rats.

Automated radiocarbonylation under the SAW conditions developed by Del Vecchio et al. [25] afforded ready-to-inject [^11^C]domperidone (5.9 ± 0.3 GBq) within 30 min in 47 ± 4% RCY and 85 ± 15 GBq/µmol MA (*n* = 12) (Figure 1B). Quality control revealed a chemical and RCP above 99%, making [^11^C]domperidone suitable for in vivo injection in rats.

### 3.2. In Vitro Experiments

The in vitro P-gp inhibitory potency of TQD was tested in P-gp-overexpressing cells using [^11^C]domperidone as substrate probes. The fitted concentration-inhibition curves for each radiotracer are shown in Figure 2. The estimated IC_50_ of TQD was 221 nM (198–248 nM) for [^11^C]domperidone compared to 4 nM (2-8 nM) for [^11^C]metoclopramide. The CI_95%_ of the IC_50_ values of the two tested substrates did not overlap, suggesting that each substrate’s sensibility to P-gp inhibition was significantly different (Figure 2). Mean uptake ratios were 2.6 ± 0,12 for [^11^C]domperidone. For comparison, the uptake ratio for [^11^C]metoclopramide was 1.4 ± 0.07 [24].

### 3.3. PET Imaging Experiments

Under the baseline conditions, the [^11^C]domperidone PET signal in the rat brain was lower than in surrounding tissues (Figure 3). The brain TACs are represented in Figure 4. The brain area under the curve between 0 and 30 min (AUC_brain;0__–__30_) was 3.98 ± 0.13 SUV·min. To allow for comparison of the impact of P-gp inhibition on the brain kinetics of [^11^C]domperidone with [^11^C]metoclopramide PET data, the AUC_brain;0__–__30_ of [^11^C]domperidone after 8 mg/kg of TQD injection (same TQD dose as [^11^C]metoclopramide data) was analyzed in one rat and showed no substantial increase compared to the baseline condition (AUC_brain;0__–__30_ = 4.9 SUV·min). It was then decided to increase the TQD dose to 15 mg/kg to achieve a higher level of P-gp inhibition. However, this dose did not either increase the [^11^C]domperidone brain exposure compared with baseline, which was confirmed using four animals (AUC_brain; 0__–__30_ = 4.6 ± 0.07 SUV·min, *p* < 0.05, *n* = 4) (Figure 5).

Arterial input function and metabolism were estimated in two additional rats (Figure 4 and Figure 5). [^11^C]domperidone metabolism was slower than for [^11^C]metoclopramide. TQD did not impact the fraction of parent (unmetabolized) [^11^C]domperidone. This was also observed for [^11^C]metoclopramide [21] (Figure 5).

Validation of a compartmental model or a graphical approach to describe the BBB penetration of [^11^C]domperidone did not provide reliable results. We, therefore, used the K_p,brain_ parameter, which describes the brain/plasma ratio. Baseline K_p,brain_ of [^11^C]domperidone was estimated at 2.46 ± 0.42 and was not statistically different with K_p,brain_ estimated at 1.42 ± 0.11 in the presence of P-gp inhibition. For comparison, the K_p,brain_ of [^11^C]metoclopramide in absence (K_p,brain_ = 10.12 ± 2.01) and the presence (K_p,brain_ = 46.96 ± 5.52) of P-gp inhibition were calculated. For baseline condition, the K_p,brain_ was statistically higher for [^11^C]metoclopramide than for [^11^C]domperidone (*p* < 0.05). P-gp inhibition significantly increased the K_p,brain_ of [^11^C]metoclopramide (*p* < 0.001) but did not increase the K_p,brain_ of [^11^C]domperidone (*p* > 0.05) (Figure 5). 

## 4. Discussion

PET imaging using isotopically radiolabeled drugs provides a unique method to compare their transporter-mediated brain kinetics in vivo, with translational perspectives [29]. In the present study, domperidone was radiolabeled with carbon-11 to estimate its BBB penetration in rats. Domperidone is considered an avid substrate of the P-gp [9] and may be viewed as an alternative probe to estimate the P-gp function at the BBB. The impact of P-gp inhibition on the P-gp-mediated transport of domperidone was first tested in vitro. Relevance for in vivo neuropharmacokinetics was then assessed in vivo using PET. [^11^C]domperidone PET data could then be compared with those of [^11^C]metoclopramide, which belongs to the same pharmacological class and is considered a weak substrate of the P-gp at the BBB [21].

Because metoclopramide bears a methyl group on a phenol moiety, it is a candidate for carbon-11 isotopic labeling by radiomethylation, the most standard approach to incorporate this radioisotope [30]. Radiomethylation was performed using [^11^C]methyl triflate, which afforded better yields than [^11^C]methyl iodide within shorter reaction times (2 min for [^11^C]CH_3_OTf versus 5 min for [^11^C]CH_3_I). Sodium hydroxide appeared to be the best base to perform this radiomethylation [26]. Although other solvents with higher boiling points than acetone, such as butanone, can be used to radiolabel [^11^C]metoclopramide [31], yields are comparable, and acetone offers a strong solubility power regarding the desmethyl precursor. Compared with [^11^C] metoclopramide, isotopic radiolabeling of domperidone with carbon-11 brings a real challenge. Domperidone does not offer a position for methylation, and an alternative strategy had to be developed. Domperidone displays two cyclic urea moieties, which could be radiolabeled with [^11^C]CO_2_ through carbonylation reactions either using phosgene [32], phosphazene derivatives [33], or carbon monoxide [34]. However, these methods use very reactive chemical agents and/or harsh conditions incompatible with fully functionalized molecules such as drugs. As a result, these methods were never applied to the synthesis of radiotracers for in vivo PET imaging purposes. The orthogonal SAW approach enables the isotopic radiolabeling of domperidone, among other functionalized molecules, in high yields and purity [25]. Moreover, this radiosynthesis strategy could be automated, leading to the reproducible production of [^11^C]domperidone suitable for in vivo injection in rats.

From a translational perspective, the molar activity of ready-to-inject [^11^C]domperidone allows for the injection of <100 µg of unlabeled domperidone for a classical dose of 370 MBq. This suggests that [^11^C]domperidone may be further used in humans according to the PET microdosing guidelines [35]. Moreover, preclinical data obtained in rodents showed linear pharmacokinetics, suggesting that a microdose of [^11^C]domperidone may predict the pharmacokinetics of the pharmacological dose of domperidone [36].

Brain PET images obtained after injection of [^11^C]domperidone in rats revealed its extremely low brain uptake compared with surrounding tissues, suggesting limited passage across the BBB. Baseline uptake (AUC) of [^11^C]domperidone was 2.5-fold lower (*p* < 0.0001) than the uptake of [^11^C]metoclopramide obtained in the same species and under the same conditions. However, brain PET kinetics have to be interpreted in the light of peripheral kinetics to correctly estimate the passage of radiolabeled compounds across the BBB [37]. [^11^C]Domperidone was slowly metabolized compared with [^11^C]metoclopramide. Indeed, parent (unmetabolized) [^11^C]domperidone accounted for more than >30% in plasma at 60 min post-injection, while parent [^11^C]metoclopramide could hardly be detected at this time. Interestingly, metabolites of domperidone and domperidone itself were shown to poorly cross the BBB in rats using ex vivo determination [36].

Determination of the arterial input function of [^11^C]domperidone was used to allow for kinetic modeling of the brain penetration across the BBB. However, poor estimation of kinetic parameters was obtained (data not shown). This may be linked with the negligible brain penetration of [^11^C]domperidone. Moreover, it is challenging to perform arterial blood sampling in rats during PET acquisition, and different animals were used for PET imaging and determining the arterial input function, respectively. This may complicate the kinetic modeling, especially in organs with poor uptake and rapid peak, such as the brain. Therefore, the brain penetration of [^11^C]domperidone was estimated in a model-independent manner. K_p,brain_ of [^11^C]domperidone was 2.46 ± 0.42. This is 4-fold lower than the K_p_ of [^11^C]metoclopramide, confirming a higher BBB passage for metoclopramide when P-gp is fully functional.

Then the importance of P-gp function at the BBB on the brain kinetics of [^11^C]domperidone was tested. First, the P-gp-mediated transport of [^11^C]domperidone was tested in vitro using human MDR1 gene-transfected cells. The maximal response to inhibition using TQD was 2-fold higher for domperidone compared with metoclopramide. Using a standardized method, this confirmed that domperidone is a more avid substrate of the P-gp than metoclopramide. Interestingly, testing an extensive range of concentrations of TQD, we show that high concentrations of TQD are needed to achieve maximal inhibition of P-gp in vitro. Indeed, the IC_50_ of TQD using [^11^C]domperidone as a substrate probe was 221 nM, whereas it was 4 nM when using [^11^C]metoclopramide as a probe. This IC_50_ value can be directly compared with other PET probes such as [^11^C]verapamil (45 nM) and [^11^C]N-desmethyl-loperramide (19 nM), estimated using the same experimental conditions [24]. This suggests that domperidone is refractory to P-gp inhibition in vitro and consequently that inhibition of the P-gp-mediated transport of domperidone may be challenging to achieve in vivo.

Consistently, the maximal dose of TQD used for the [^11^C]metoclopramide study (8 mg/kg) did not impact the brain kinetics and BBB penetration of [^11^C]domperidone in rats (*n* = 1). A higher dose was therefore tested (15 mg/kg, *n* = 4) and did not either enhance the brain penetration of [^11^C]domperidone in vivo (*p* > 0.05). Using in situ brain perfusion, Dagenais and colleagues reported a 3.1-fold higher passage of domperidone in P-gp-deficient mice than in wild-type animals [38]. Moreover, a massive dose of the P-gp inhibitor cyclosporine (2.5 µM in brain perfusion) increased the brain penetration of [^11^C]domperidone by 1.87-fold in mice, which enhanced the catalepsy induced by a pharmacological dose of domperidone [19]. Although species differences in the P-gp-mediated substrates between mice and rats may occur [39], it is likely that the high dose of TQD administered in our study was insufficient to achieve complete inhibition of the P-gp-mediated transport of domperidone in rats.

An explanation for how P-gp can maintain the interaction with domperidone after blocked activity by TQD is the interaction of domperidone with a different biding site of P-gp than TQD. However, TQD seems to have an action on the P-gp conformation and the ATPase activity [40] in a fixation site-independent manner even if a biding site interaction cannot be totally excluded without further investigations [41]. Another possibility is the interaction of domperidone with other efflux transporters at the BBB other than P-gp.

From a clinical perspective, complete P-gp inhibition leading to the situation of knockout animals is rarely achieved [42]. This especially holds for domperidone, which behaves as an inhibition-refractory substrate compared with metoclopramide or other substrates [24]. This suggests that domperidone shows a limited passage across the BBB compared with metoclopramide. Moreover, our in vitro assay using cells expressing the human P-gp and in vivo PET experiments performed in rats suggest that inhibition of the P-gp mediated transport of domperidone at the BBB is very unlikely. This supports the use of domperidone for pathological conditions where CNS effects of D2 antagonists have to be avoided, such as the prevention of emesis induced by DOPA therapy. Moreover, our study does not support a risk for DDIs involving P-gp at the BBB between domperidone and co-administered drugs.

## 5. Conclusions

From an imaging perspective, [^11^C]domperidone was initially evaluated as an avid substrate probe for PET imaging of P-gp function at the BBB. However, [^11^C]domperidone is poorly sensitive to changes in P-gp function induced by TQD, a potent and clinically validated inhibitor of P-gp at the BBB. [^11^C]domperidone is therefore not a suitable radiotracer to assess the importance of P-gp activity at the BBB.

From a neuropharmacology perspective, the direct comparison of the brain kinetics of [^11^C]domperidone and [^11^C]metoclopramide illustrates the potential of PET to evaluate the cerebral passage of two molecules of the same family, which could be extended to two drug candidates with a phase 0 design considering the inter-species differences.

## Figures and Tables

**Figure 1 pharmaceutics-14-01658-f001:**
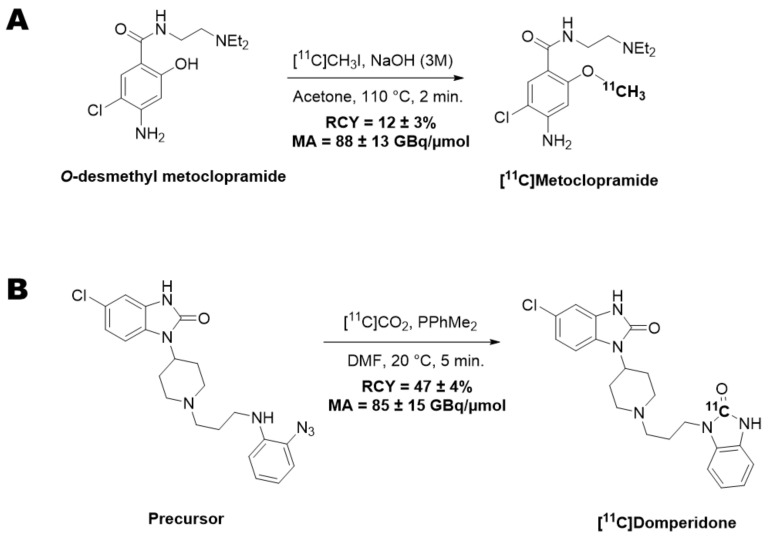
Radiosynthesis scheme of [^11^C]metoclopramide (**A**) and [^11^C]domperidone (**B**).

**Figure 2 pharmaceutics-14-01658-f002:**
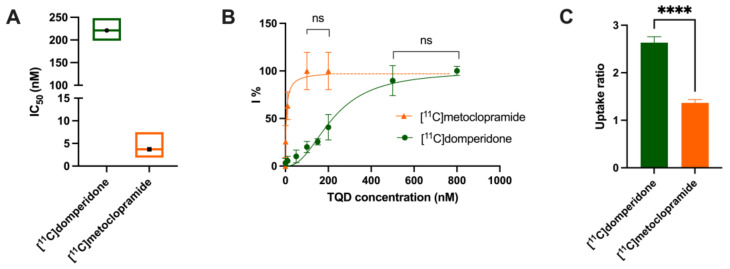
In vitro transport data. Comparison of the in vitro half-maximum inhibitory concentration (IC_50_) (**A**) and uptake ratio (**C**) of [^11^C]domperidone and [^11^C]metoclopramide assessed using increasing doses of tariquidar (TQD) in MDCKII-MDR1 cells. In (**B**), concentration-inhibition curves (I%, extent of inhibition in percent) are shown. In (**A**), data are shown as mean ± CI_95%_ (95% confidence interval) with *n* = 4 per condition. In (**B**,**C**), data are shown as mean ± S.D with *n* = 4 per condition. Lines in (**B**) represent fits of the employed Hill model. **** *p* < 0.0001, ns = non-significant (*t*-test for comparison).

**Figure 3 pharmaceutics-14-01658-f003:**
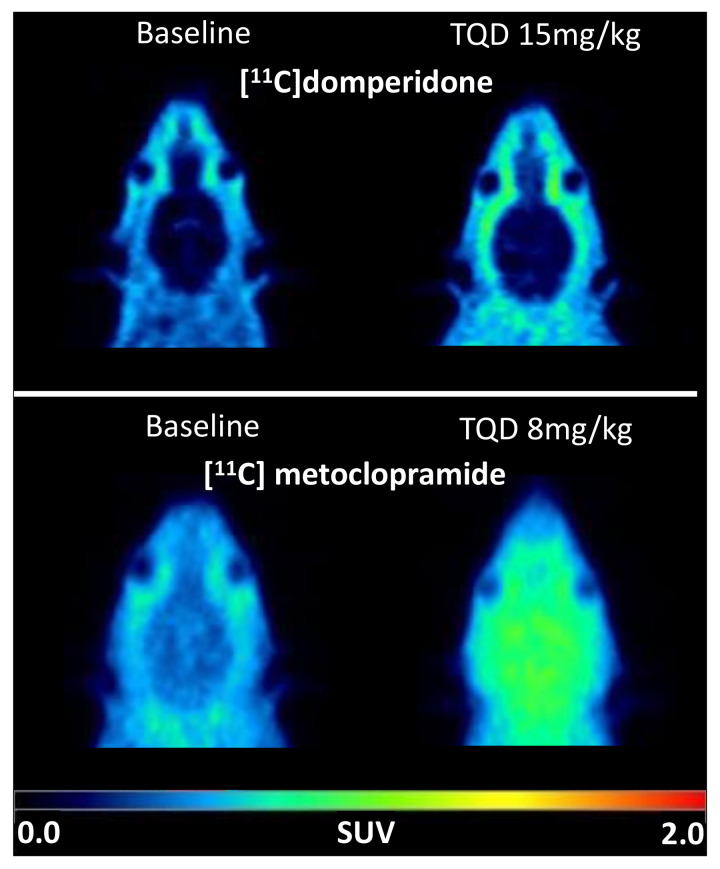
PET images were obtained with [^11^C]domperidone and [^11^C]metoclopramide after selected doses of the P-gp inhibitor tariquidar (TQD). Representative PET images are summed from 0 to 30 min and were obtained at baseline and after P-gp inhibition (8 mg/kg TQD for [^11^C]metoclopramide and 15 mg/kg TQD for [^11^C]domperidone).

**Figure 4 pharmaceutics-14-01658-f004:**
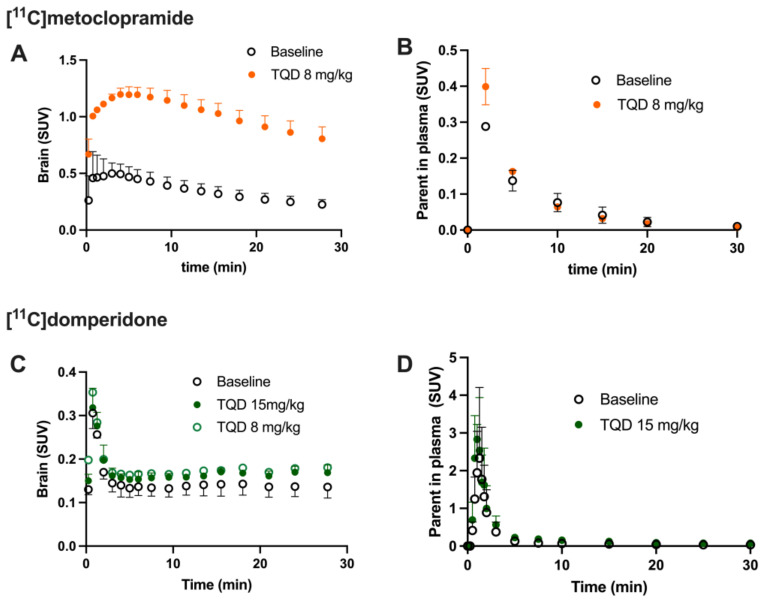
Impact of P-gp inhibition of the brain kinetics of [^11^C]domperidone compared to [^11^C]metoclopramide. Time-activity curves (TAC) were obtained with [^11^C]metoclopramide (**A**,**B**) and [^11^C]-domperidone (**C**,**D**) in the absence and presence of P-gp inhibitor tariquidar (TQD). (**A**,**C**) present the raw data in the brain, while (**B**,**D**) are the corresponding plasma TAC curves corrected from metabolism exposures. Data are reported as mean ± S.D with *n* = 4 per brain condition and *n* = 2 per plasma condition.

**Figure 5 pharmaceutics-14-01658-f005:**
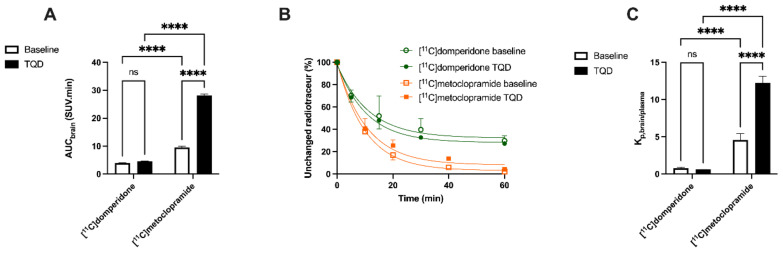
Impact of P-gp inhibition of brain exposure and metabolism of [^11^C]domperidone compared to [^11^C]metoclopramide. In (**A**), brain exposures were obtained with [^11^C]domperidone and [^11^C]metoclopramide (AUC_brain_) in the presence and absence of P-gp inhibitor tariquidar (TQD). In (**B**), respective parent fraction in plasma of radiotracer in same conditions. In (**C**), the corresponding K_p,brain/plasma_ are shown. Data are reported as mean ± S.D with *n* = 4 per condition. **** *p* < 0.0001, ns = non-significant (two-way ANOVA with Tukey’s post hoc test for multiple comparison).

**Table 1 pharmaceutics-14-01658-t001:** Comparison of neurological side effects between domperidone and metoclopramide [4,5,6].

Side Effects	Very Frequent	Frequent	Not Frequent	Rare	Unspecified
Domperidone 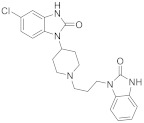			dizziness, sleepiness, headache, extrapyramidal signs		restless legs syndrome
Metoclopramide 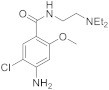	sleepiness	extrapyramidal signs, parkinsonian syndrome, akathisia	dystonia, dyskinesia, altered consciousness	convulsions	late dyskinesia, neuroleptic malignant syndrome

## Data Availability

Data are freely available upon request to the corresponding author.

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
