# Peer review of "Comparison of the Blood–Brain Barrier Transport and Vulnerability to P-Glycoprotein-Mediated Drug–Drug Interaction of Domperidone versus Metoclopramide Assessed Using In Vitro Assay and PET Imaging"

_pharmaceutics, 2022, doi:10.3390/pharmaceutics14081658_

Round 1
Reviewer 1 Report
This article demonstrated the different behavior of two drugs for dopamine D2 receptors toward p-glycoprotein using in vitro assay and in vivo PET imaging studies. The results are interesting and might be good examples of pharmaceutical research using PET.
However, the reviewer requests more explanation why domperidone does not penetrate the brain despite the blocking conditions of p-glycoprotein. In the p-glycoprotein experiment, TQD was injected 15 min prior to the radiotracer injection. Therefore, TQD binding to the p-glycoprotein was not affected by radiotracer injection. The authors explained that domperidone is an avid p-glycoprotein substrate. They did not explain how p-glycoproteins can maintain the interaction with domperidone even after their activity is blocked by TQD.
Therefore, the reviewer suggests two possibilities to explain the phenomena in the Discussion. Domperidone might interact with different binding sites of p-glycoprotein. Domperidone might interact with different efflux proteins other than p-glycoproteins. The authors can run additional experiments or literature searches based on these hypotheses. If they can explain different ways, please do so.
The reviewer also asks to fix the following minor errors.
1. The authors use 20% ethanol in saline for the final reformulation. Is this acceptable for the animal experiment?
2. Lines 122 and 142: Molar activity has to be expressed at a specific time point such as the EOS. Please write what specific time point you measured and calculated.
3. Line 135: The information of the Semi-preparative column is not correct. Please check again. Does Agilent sell Symmetry C18 column? 4.6 mm Diameter correct?
4. Line 157 and 158: Did you calculate RCY using the radioactivity at the same time point? The equation looks like decay-uncorrected RCY.
5. In Figure 1A. Please check the unit of MA.
6. Line 209 à Is this section title? If so, please write in italic.
7. Lines 122, 185, 204, 211, 280, 372: minutes à min. You also used min instead of minutes in the main text many times. Please be consistent
8. You can use TQD instead of tariquidar because you showed tariquidar (TQD).
9. Line 244 à Radio activity should be one word.
10. Line 127. CO2 à CO2
11. Line 229 and 230: trifluoroacetic acid à TFA because you used TFA previously.
12. Lines 270, 309, 342, 344: time-activity curve à TAC because you used TAC previously.
13. Line 272: “was” should be written in normal size.
14. Lines 282, 286: radiochemical purity à RCP because you used RCP previously.
15. Line 388: molar activity à MA because you used MA previously.
16. Line 414: 11C à 11C
17. Line 438, 444: CsA and KO à Please write the definition.
Author Response
We thank the reviewers for their comments. We have addressed all points they have raised and changed the manuscript accordingly.
Referee: 1
This article demonstrated the different behavior of two drugs for dopamine D2 receptors toward p-glycoprotein using in vitro assay and in vivo PET imaging studies. The results are interesting and might be good examples of pharmaceutical research using PET.
However, the reviewer requests more explanation why domperidone does not penetrate the brain despite the blocking conditions of p-glycoprotein. In the p-glycoprotein experiment, TQD was injected 15 min prior to the radiotracer injection. Therefore, TQD binding to the p-glycoprotein was not affected by radiotracer injection. The authors explained that domperidone is an avid p-glycoprotein substrate. They did not explain how p-glycoproteins can maintain the interaction with domperidone even after their activity is blocked by TQD.
Therefore, the reviewer suggests two possibilities to explain the phenomena in the Discussion. Domperidone might interact with different binding sites of p-glycoprotein. Domperidone might interact with different efflux proteins other than p-glycoproteins. The authors can run additional experiments or literature searches based on these hypotheses. If they can explain different ways, please do so.
Thank you for the suggestion. We have accordingly added a paragraph in the discussion section. Please see lines 547-552.
The reviewer also asks to fix the following minor errors.
- The authors use 20% ethanol in saline for the final reformulation. Is this acceptable for the animal experiment?
The final formulation at the end of the production indeed contains 20% ethanol in saline, but this solution is further diluted in saline before injection to inject less than 10% ethanol. We have added a sentence to clarify this point, lines 292-293.
Lines 122 and 142: Molar activity has to be expressed at a specific time point such as the EOS. Please write what specific time point you measured and calculated.
The molar activity is expressed at the end of the beam (EOB), and the specification has been added to lines 195 and 214.
- Line 135: The information of the Semi-preparative column is not correct. Please check again. Does Agilent sell Symmetry C18 column? 4.6 mm Diameter, correct?
Indeed, Symmetry columns are manufactured by Waters. The text has been changed accordingly line 207.
- Line 157 and 158: Did you calculate RCY using the radioactivity at the same time point? The equation looks like decay-uncorrected RCY.
All activities have been decay-corrected. It is now clearly written on lines 229-230.
- In Figure 1A. Please check the unit of MA.
MA is expressed in GBq/µmol. The typo in Figure 1A has been corrected.
- Line 209 à Is this section title? If so, please write in italic.
Corrected, thank you.
- Lines 122, 185, 204, 211, 280, 372: minutes à min. You also used min instead of minutes in the main text many times. Please be consistent
Corrected, thank you.
- You can use TQD instead of tariquidar because you showed tariquidar (TQD).
Corrected, thank you.
- Line 244 à Radio activity should be one word.
Corrected, thank you.
- Line 127. CO2 à CO2
Corrected, thank you
- Line 229 and 230: trifluoroacetic acid à TFA because you used TFA previously.
Corrected, thank you.
- Lines 270, 309, 342, 344: time-activity curve à TAC because you used TAC previously.
Corrected, thank you.
- Line 272: “was” should be written in normal size.
Corrected, thank you.
- Lines 282, 286: radiochemical purity à RCP because you used RCP previously.
Corrected, thank you.
- Line 388: molar activity à MA because you used MA previously.
Corrected, thank you
- Line 414: 11C à 11C
Corrected, thank you
- Line 438, 444: CsA and KO à Please write the definition.
Replaced by full names “cyclosporin A” and “knockout”, thank you.
Reviewer 2 Report
This manuscript by Breuil et al. reports the comparison of the blood-brain barrier transport and vulnerability to P-glycoprotein-mediated drug-drug interaction of domperidone versus metoclopramide assessed using in vitro assay and PET imaging. This is an interesting study, I recommend a minor revision.
1. More background should be included in "INTRODUCTION".
2. Characterization data of 11C metoclopramide and 11C domperidone should be added.
3. It is suggested to add a scheme describing the main content of this study at the beginning of the manuscript.
Author Response
We thank the reviewers for their comments. We have addressed all points they have raised and changed the manuscript accordingly.
This manuscript by Breuil et al. reports the comparison of the blood-brain barrier transport and vulnerability to P-glycoprotein-mediated drug-drug interaction of domperidone versus metoclopramide assessed using in vitro assay and PET imaging. This is an interesting study, I recommend a minor revision.
- More background should be included in "INTRODUCTION".
We have added a paragraph to describe the available background regarding the comparison of in vitro and in vivo data for the brain kinetics of domperidone vs metoclopramide. Please see lines 76-134.
- Characterization data of 11C metoclopramide and 11C domperidone should be added.
We have added the HPLC and radioHPLC data in supplemental data to better report the characterization of [11C]metoclopramide and [11C]domperidone.
- It is suggested to add a scheme describing the main content of this study at the beginning of the manuscript.
A graphical abstract has been added at the beginning of the manuscript.